# Oocyte Selection for In Vitro Embryo Production in Bovine Species: Noninvasive Approaches for New Challenges of Oocyte Competence

**DOI:** 10.3390/ani10122196

**Published:** 2020-11-24

**Authors:** Luis Aguila, Favian Treulen, Jacinthe Therrien, Ricardo Felmer, Martha Valdivia, Lawrence C Smith

**Affiliations:** 1Centre de Recherche en Reproduction et Fértilité (CRRF), Université de Montréal, St-Hyacinthe, QC J2S 2M2, Canada; jacinthe.therrien@umontreal.ca (J.T.); lawrence.c.smith@umontreal.ca (L.C.S.); 2School of Medical Technology, Faculty of Science, Universidad Mayor, Temuco 4801043, Chile; faviantreulen@gmail.com; 3Laboratory of Reproduction, Centre of Reproductive Biotechnology (CEBIOR-BIOREN), Faculty of Medicine, Universidad de La Frontera, Temuco 4811322, Chile; ricardo.felmer@ufrontera.cl; 4Laboratory of Animal Reproductive Physiology, Biological Sciences Faculty, Universidad Nacional Mayor de San Marcos, Lima 15088, Peru; mvaldiviac@unmsm.edu.pe

**Keywords:** oocyte competence, livestock production, assisted reproductive technology, embryo development, micromanipulation, in vitro production

## Abstract

**Simple Summary:**

The efficiency of producing embryos using in vitro technologies in cattle species remains lower when compared to mice, indicating that the proportion of female gametes that fail to develop after in vitro manipulation is considerably large. Considering that the intrinsic quality of the oocyte is one of the main factors affecting embryo production, the precise identification of noninvasive markers that predict oocyte competence is of major interest. The aim of this review was to explore the current literature on different noninvasive markers associated with oocyte quality in the bovine model. Apart from some controversial findings, the presence of cycle-related structures in ovaries, a follicle size between 6 and 10 mm, a large slightly expanded investment without dark areas, large oocyte diameter (>120 microns), dark cytoplasm, and the presence of a round and smooth first polar body have been associated with better embryonic development. In addition, the combination of oocyte and zygote selection, spindle imaging, and the anti-Stokes Raman scattering microscopy together with studies decoding molecular cues in oocyte maturation have the potential to further optimize the identification of oocytes with better developmental competence for in vitro technologies in livestock species.

**Abstract:**

The efficiency of producing embryos using in vitro technologies in livestock species rarely exceeds the 30–40% threshold, indicating that the proportion of oocytes that fail to develop after in vitro fertilization and culture is considerably large. Considering that the intrinsic quality of the oocyte is one of the main factors affecting blastocyst yield, the precise identification of noninvasive cellular or molecular markers that predict oocyte competence is of major interest to research and practical applications. The aim of this review was to explore the current literature on different noninvasive markers associated with oocyte quality in the bovine model. Apart from some controversial findings, the presence of cycle-related structures in ovaries, a follicle size between 6 and 10 mm, large number of surrounding cumulus cells, slightly expanded investment without dark areas, large oocyte diameter (>120 microns), dark cytoplasm, and the presence of a round and smooth first polar body have been associated with better competence. In addition, the combination of oocyte and zygote selection via brilliant cresyl blue (BCB) test, spindle imaging, and the anti-Stokes Raman scattering microscopy together with studies decoding molecular cues in oocyte maturation have the potential to further optimize the identification of oocytes with better developmental competence for in-vitro-derived technologies in livestock species.

## 1. Introduction

In recent years, new knowledge in the field of assisted reproductive technologies (ART, has allowed researchers and practitioners to reach new hallmarks in oocyte and sperm in vitro competence. Gamete competence is the ability to undergo successful fertilization and develop a normal blastocyst that is capable of implanting in the uterus and generate viable offspring [1]. Many researchers are focused on identifying cellular and molecular markers to select the most competent oocyte and spermatozoon to produce embryos with higher implantation potential [2].

Although it is well known that the most common applications of ARTs in livestock species are for research purposes, some techniques, particularly in vitro embryo production (IVP), have become commercially viable and are extensively used for animal breeding [3]. Nonetheless, the efficiency of IVP technologies in livestock species, such as bovine, equine, and porcine, measured as the proportion of immature oocytes that reach the blastocyst stage, rarely exceeds the 30–40% threshold [4], which means that the proportion of oocytes that fail to develop following in vitro maturation, fertilization, and culture is considerably large. Contrary to humans, where eggs are mainly collected at the MII stage, in livestock species, the oocytes have to be matured in vitro due to the difficulty of obtaining a sufficient number of in vivo matured oocytes [5]. Additionally, given that the most frequent source of ovaries is slaughterhouse-derived animals, many important factors that influence oocyte quality, such as age of the donor, the stage of the estrous cycle, nutritional status, genetic potential, presence of a reproductive disorder, and others, are often unknown [6]. Therefore, it is almost impossible to avoid the retrieval of a heterogeneous population of oocytes that have a distinct ability to undergo maturation and support early embryonic development after fertilization, which is known as developmental competence or oocyte quality [7].

Considering that the intrinsic quality of the oocyte is one of the major factors affecting early embryonic development [8], and that embryo culture conditions have a crucial role in determining blastocyst quality [9], the precise selection of competent oocytes is vital for IVP technologies in livestock. Recently, the new arrival of bovine embryonic stem cells (ESCs) [10,11] emphasizes the already existing challenge in the selection of competent oocytes for the production of high-quality embryos through in vitro fertilization (IVF), intracytoplasmic sperm injection (ICSI) or somatic cell nuclear transfer (SCNT), and derivation of pluripotent stem cell lines, with promising applications in research or industry, such as in vitro breeding programs [12]. Usually, for IVP and micromanipulation procedures (ICSI and SCNT), the choice of the oocytes lie in morphological features that are easily assessed with light microscopy [13]. The major difference and/or advantage of conventional IVF compared to micromanipulation procedures is that fertilization can occur during gamete co-incubation when the oocyte has reached or is close to nuclear and cytoplasmic maturity [14]. Conversely, during micromanipulation procedures, the operator must accurately assess the maturity of the oocyte and, therefore, its competence [15]. Because the criteria used for grading and selecting oocytes vary among researchers, could be easily misinterpreted, and depend on the expert’s evaluation and experience, the identification of noninvasive cellular or molecular markers that predict oocyte competence is a major research goal [16,17]. Despite efforts for finding molecular factors associated with oocyte quality, it is still challenging to find a visual marker that accurately predicts embryonic competence. Thus, this article reviews the current literature on different noninvasive markers that have been correlated with oocyte quality in cattle and explores the utility of each grading system.

## 2. Morphological and Visual Markers for the Selection of the Best Oocytes

### 2.1. Ovarian Morphology

During the retrieval of oocytes from slaughterhouse material, the collection of ovaries based on the presence or absence of estrus cycle structures, i.e., presence or absence of follicles and corpus luteum (CL), has been used as a straightforward noninvasive criterion to access developmentally competent oocytes. However, there are discrepancies among different studies in this regard. Early studies indicated that the presence of a dominant follicle (>10 mm) in one or both ovaries had a negative effect on in vitro developmental competence of oocytes derived from the subordinate follicles [18,19,20]. Manjunatha et al. [21] reported that embryonic development was higher in oocytes coming from ovaries with a CL and no dominant follicle, whereas gametes coming from ovaries that had a CL and a dominant follicle showed higher competence only when oocytes were derived from the dominant follicle. In agreement with this notion, Pirestani et al. [22] reported that oocytes derived from ovaries containing a large follicle (~20 mm) were less competent compared to those derived from ovaries containing a CL. Similarly, Penitente-Filho et al. [23] classified cumulus–oocyte complexes (COCs) under the stereomicroscope and indicated that ovaries with CL yielded a larger number of competent oocytes than ovaries without CL. However, the oocytes used in the latter study were not subjected to IVP to confirm their developmental competence. Overall, these studies indicate that the presence of a dominant follicle in the bovine ovary would negatively influence the subsequent embryo development, while the presence of a CL favors oocyte competence. In contrast, more recent studies indicated that the presence of a CL has negative effects on the developmental competence of ipsilateral oocytes [24,25]. However, this “negative” effect does not influence the competence of oocytes originated from large follicles (10–20 mm) as much as those derived from small and medium follicles (<9 mm) [25].

Ovaries without structures indicative of estrus cyclicity have less competent oocytes than others [21,26], as indicated by the presence of fewer than 10 follicles 2–5 mm in diameter and no large follicles [27]. In addition, other authors indicated that the developmental competence of bovine oocytes from antral follicles (2 to 8 mm) is not affected by either the presence of a dominant follicle or the phase of folliculogenesis [27,28,29,30,31]. Thus, despite the few discrepancies, it seems that the selection of ovaries based on the presence of cycle-related structures could help optimize access to oocytes with better developmental competence for in-vitro-derived technologies. Nevertheless, the positive or negative effects of ovarian structures on oocyte competence require further investigation to determine more precisely how these ovarian structures impact subsequent in vitro embryonic development.

### 2.2. Follicle Size

One of the most used criteria to obtain competent oocytes is the size of the follicle. Research over the past decades indicates that bovine oocytes gain competence at late stages of the follicular phase, when signs of atresia are observed for the first time, such as a slight expansion in the outer cumulus layers and some cytoplasmic granulations [7,32]. Therefore, the recommendation is that oocytes recovered from follicles between 6 and 10 mm develop more frequently to more advanced embryonic stages [7,33,34,35,36]. Although the acquisition of competence begins when the follicle reaches 3 mm and the effect of size becomes more important at 8 mm [19,37,38], success is not guaranteed even if the oocytes come from larger follicles [39].

The acquisition of oocyte competence seems to be due to the substrate support received and to the developmental phase at the time of removal from the follicle [7,32,34]. Recent reports indicate that the follicular fluid (FF) microenvironment of large follicles has higher levels of electrolytes, glucose, reactive oxygen species, glutathione, superoxide dismutase activity, lipids, cholesterol, pyruvate, and estradiol [33,40,41]. Moreover, oocytes derived from larger follicles also show a different transcriptional pattern for chromatin remodeling and metabolic pathways, such as lipid metabolism, cellular stress, and cell signaling, with respect to those coming from smaller sizes, which would favor their developmental potential [41,42]. Therefore, these findings indicate that large follicles (>6 mm) provide an appropriate microenvironment for the oocyte leading to better embryonic development.

### 2.3. Morphology of the Cumulus–Oocyte Complexes

The quality of COCs can be influenced by multiple factors, both intrinsic and extrinsic. Intrinsic factors include breed, age, reproductive status, metabolic and nutritional status, hormonal levels, and stage of the estrous cycle [43], whereas key extrinsic factors include the timing between slaughter and oocyte withdrawal from the ovary, morphology and methods of collecting the COCs, storage temperature of the ovaries, collection media, and micromanipulation skills of the operator [44].

Since intrinsic factors are more difficult to control when using slaughterhouse ovaries from cows of unknown origin, the morphology of the COC is relatively easy to evaluate and is often the most common criterion used to select and classify a standard collection of bovine oocytes [45,46,47]. Morphological criteria include the number and appearance of cumulus layers and the cytoplasmic features of the oocyte, such as the texture or brightness of its cytoplasm. Basically, the healthiest COC quality (Class I) relates to a complete cumulus cover with several compact cell layers; medium quality (Class II) has only partial cumulus cover and/or slightly expanded cumulus containing fewer than five cell layers; lastly, the worst quality (Class III) has a darker cytoplasm and the presence of dark spots with expanded cumulus, all indicative of follicular atresia (Figure 1). However, such classification criteria vary among laboratories.

The study by Wit et al. [30] classified COCs into three groups: (i) compact and bright, (ii) less compact and dark, and (iii) strongly expanded cumulus with dark spots, where developmental capacity, measured by in vitro embryo production, was correlated with COC appearance. Moreover, less compact and darker COCs showed faster meiotic resumption. Another study using similar categories reported that COCs with darker cumulus and ooplasm were the most competent in terms of cleavage and blastocyst yield after IVF and parthenogenetic activation [48]. In addition, this study showed that developmental competence was related to calcium currents in the plasma membrane and calcium stores in the cytoplasm of immature oocytes [48]. The report by Bilodeau-Goeseels et al. [49] divided COCs into six classes on the basis of their cumulus and ooplasm features. These authors found that, although oocytes with fewer than five layers of cumulus cells (CC) showed lower cleavage rates, their developmental potential to the blastocyst stage was similar to oocytes with more than five layers of CC. More recently, De Bem et al. [37] found that class III COCs, considered to be of poor morphological quality, were superior in terms of blastocyst development to the intermediate class II group, but similar to class I COCs, albeit without differences in blastocyst quality. Emanuelli et al. [50] indicated that COCs with partial (fewer than five cell layers) and expanded cumulus had higher levels of DNA fragmentation after in vitro maturation (IVM) and lower competence compared to healthier ones, in accordance with the report by Yuan et al. [51]. However, blastocysts derived from COCs with varied morphologies exhibited no variations in terms of quality assessed by the number of cells. In addition, Emanuelli et al. [50] further concluded that these differences were due to better nuclear maturation through enhanced maintenance of metaphase II (MII) block by COCs showing full cumulus coverage.

Thus, despite these contradictory results, most studies agree that COCs showing signs of early atresia yield high blastocyst rates compared to morphologically healthy COCs. Nonetheless, advanced atresia, with signs such as cytoplasmic granulations, fewer than five cumulus layers, and expanded cumulus with dark cellular masses or, strictly, its complete absence, show lower in vitro potential as measured by cleavage rates and blastocyst formation [30] (Figure 1C). Additionally, although morphological classification seems to influence the proportion of blastocysts formed, such criteria may not influence their quality. Therefore, when selecting COCs according to their cumulus investment and ooplasm texture, the ideal would be to target COCs with several cumulus cell layers (more than or at least five layers), compact and/or slightly expanded, with or without dark areas in the oocyte and cumulus.

### 2.4. Lipid Content

The morphological appearance of the ooplasm commonly assessed to select the oocytes [52,53] is influenced by lipid content in livestock species, such as cattle, pigs, and horses [54,55,56]. Lipids, in the form of lipid droplets (LDs), are signaling molecules with important roles in oocyte maturation and competence acquisition [57]. In the late stage of oocyte maturation and during preimplantation development, endogenous oocyte lipids work as an energy source [58,59] and as a lipid factory for energy reserve [60]. Failure to use lipids in oocytes has been shown to be related to inadequate nuclear maturation [61,62]. The number of LDs present in the cytoplasm increases as the oocyte grows [63] and, although the ooplasm organization does not undergo major changes during in vitro maturation to MII [56], the type and number of lipids in the LDs seem to be more dynamic and to undergo changes during meiotic progression to MII [59,64].

LDs aggregate in the form of dark clusters that can be seen in the ooplasm as a cytoplasmic darkness [55,65] (Figure 2). Cytoplasmic darkness can be homogeneous, affecting the entire cytoplasm or concentrated in the center, with a clear peripheral ring that gives the cytoplasm a darkened appearance (Figure 2B,D). This opaque appearance is more intense in pigs and domestic cats, followed by cows and finally sheep and goats, whose ooplasm is lighter. In the case of horses, lipid polarization is commonly observed, which facilitates the visualization of the spermatozoon within the oocyte [55,66].

Several studies investigated the relationship between oocyte lipid content and competence. For instance, cytoplasm color can be used as a marker of lipid content and as predictive of the embryonic potential [67], as oocytes with a uniform and brown or dark cytoplasm contain more intracellular lipids than oocytes with a granular or pale cytoplasm [65]. Most studies demonstrated that oocytes with rough granulations or very pale ooplasm yield a lower preimplantation development [49,53,67]. Jeong et al. [68] classified the ooplasm in three categories: dark, brown, and pale. In this study, the content of mitochondria and the proportion of oocytes that reached the blastocyst stage were higher in darker oocytes. Moreover, Nagano et al. [67] reported that sperm penetration, monospermic fertilization, cleavage, and blastocyst rates were higher in oocytes with a brown ooplasm compared to those with pale or very dark ones. Moreover, brown oocytes with a dark edge or with dark spots showed, under electron microscopy, an organelle arrangement similar to in vivo matured oocytes, and pale or black oocytes appeared to be degenerating and/or aging [67]. The authors concluded that a dark ooplasm is associated with a lipid accumulation and better developmental competence, while a pale ooplasm would indicate fewer organelles and poor developmental potential [69]. Interestingly, a study by Prates et al. [70] distinguished fat areas of different color shades using the Nomarski interference differential contrast (NIC) as the fat gray value of porcine oocytes, reflecting alterations in lipid content, and proposed this tool as an appropriate and noninvasive technique to evaluate the lipid content of a single oocyte before or after in vitro maturation. Recently, the study of Jasensky et al. [71] reported the use of anti-Stokes Raman scattering (CARS) microscopy as a new non-invasive tool for the quantification of lipid content in mammalian oocytes. This study showed that the ~2 min of laser exposure was enough for a quantitative comparison of lipid content in mice oocytes at different developmental stages, as well as in oocytes of others mammalian species, and, more importantly, without detrimental effects (without the need to attach fluorescence labels) for subsequent preimplantation development. Thus, its application in live-cell imaging of oocytes is promising to provide alternative and/or additional information in order to improve the accuracy of subjective morphometric measurements.

Taken together, as stated by the review of Nagano and colleagues [69], a dark ooplasm indicates an accumulation of lipids and good developmental potential, a light-colored ooplasm indicates a deficiency of lipid stores and poor developmental potential, and a black ooplasm indicates aging and low developmental potential (Figure 2). Finally, the use of NIC and CARS should be further investigated as a potential noninvasive tool to evaluate the lipid content of single oocytes in livestock species.

### 2.5. Cumulus Expansion and Oocyte Size

Another parameter that is often used as an indirect indicator of oocyte quality is the degree of cumulus expansion following maturation, typically after 20 to 24 h of culture in an in vitro maturation environment. Grades 1 to 3 (sometimes 4) are attributed to increasing degrees of expansion (1: modest expansion, characterized by few morphologic changes compared to before maturation, 2: partial expansion, and 3: complete or almost complete expansion) [72,73,74].

Although the expansion of CCs has been described as the basis for oocyte maturation [75] and early reports supported the idea that quantity and quality of the expanded cumulus mass were correlated with developmental capacity [76], its usefulness as an indicator of developmental potential in bovine seems to be modest [77]. For instance, studies by Anchordoquy et al. [78], Dovolou et al. [79], and Rosa et al. [80] reported that, under different experimental conditions, the cumulus expansion index was not indicative of blastocyst yield or quality. Similarly, another study indicated that inhibition of cumulus expansion by enzymatic hyaluronidase degradation did not affect cleavage or blastocyst development [81]. Nonetheless, as shown by Fukui et al. [82], more than an indicator of developmental competence, CCs and their expansion play an important role in fertilization by inducing the acrosome reaction and, therefore, promoting higher fertilization rates.

In addition to follicle size, oocyte size has been used as a noninvasive quality parameter. Although it is difficult to measure the precise diameter of the oocyte during IVF, oocyte selection based on diameter can be used as a routine step during micromanipulation protocols. The study of Fair et al. [83] classified oocytes recovered from slaughterhouse ovaries into four groups (<100 microns, 100 to 110 microns, 110 to 120 microns, and >120 microns). Rates of resumption of meiosis to MII were higher for oocytes >110 microns. Moreover, oocytes <110 microns were transcriptionally active, suggesting that they were still in the growth phase of oogenesis [83,84]. Similarly, Anguita et al. [85] reported that cleavage and blastocyst rates were higher in oocytes >110 microns. Moreover, Otoi et al. [86] and Arlotto et al. [29] found that oocytes >115 microns had better rates of nuclear maturation and a lower incidence of polyspermy after IVF, but cleavage rates and development to the blastocyst stage were optimal in oocytes >120 microns. Huang et al. [87] and Yang et al. [88] compared oocytes collected from initial antral follicles (0.5–1 mm in diameter) cultured in vitro for 14–16 days with oocytes collected from antral follicles (2–8 mm in diameter), cultured, and submitted to IVM. The authors reported better maturation rate for oocytes >115 microns, optimal for oocytes >120 microns, but developmental competence was only high for oocytes collected from antral follicles and of size >120 microns.

These results suggest that bovine oocytes acquire meiotic competence with a diameter of 115 microns, but full developmental competence is acquired around 120 microns, possibly because smaller oocytes have not yet completed their growth phase [46]. Thus, the selection of follicles between 6 and 10 mm, with oocyte diameters >115 and <130 microns, has the potential to optimize developmental outcomes.

### 2.6. First Polar Body Assessment

At the end of IVM and after the removal of CCs, it is easy to perform a detailed observation of morphological features [13], including the assessment of oocyte shape, cytoplasm color and granulation, regularity and thickness of the zona pellucida, size of the perivitelline space, presence of vacuoles, and presence or absence of the first polar body (PB1) and its morphology. Extrusion of PB1 in mammalian oocytes is a cellular landmark of meiotic maturation, and its assessment is frequently used as an indicator of nuclear maturation [89]. Thus, its absence indicates that the oocyte is immature or that it has degraded due to aging; however, its presence does not guarantee that the oocytes have completed their maturation process, and some of them remain incompetent despite exhibiting morphologic features of nuclear maturation [90].

In bovine species, extrusion of PB1 begins at 16–18 h after IVM [91,92,93,94]. Nonetheless, oocytes acquire the highest developmental competence at around 5–10 h after PB1 extrusion [14,95]. Dominko and First [95] indicated that oocytes that extruded their PB1 after 16 h of IVM were only capable of reaching higher developmental competence after 24 h of in vitro culture. Thus, cytoplasmic maturation in cattle occurs several hours after nuclear maturation, probably between 24 and 30 h after the beginning of IVM.

Unfortunately, there are no studies that analyzed the influence of the first PB morphology on oocyte competence in cattle. However, one study using porcine oocytes indicated that PB1 with a smooth or intact surface was indicative of a more advanced cytoplasmic maturation and better embryonic development in vitro than those with a fragmented or rough surface [96]. Despite lacking studies in domestic species, studies in humans investigated the association between PB1 morphology and oocyte competence [97,98]. Ebner et al. [99] conducted a retrospective study using 70 consecutive ICSI cases in which oocyte classification based on PB1 morphology revealed that oocytes with intact, well-shaped PB1 yield better fertilization and high embryonic quality. Later, Ebner et al. [97] confirmed the relationship among PB1 morphology, fertilization, and blastocyst quality, as well as a positive effect on implantation and pregnancy rates. Similarly, Rose et al. [100] reported that oocytes with an intact PB1 show better fertilization and embryonic development, whereas those displaying a PB1 with morphological abnormalities such as a larger size, irregularities, coarse surface, or fragmentation are less competent during an IVF protocol, having poor implantation capabilities after embryo transfer. In contrast, others did not report any correlation [101,102,103]. Thus, there is a lack of consensus on the impact of PB1 morphology on oocyte competence and embryonic development in humans. It is also important to note that some PB1 abnormalities may be an artefact of oocyte manipulation (mainly during the denudation process) or aging [104].

In summary, although the selection of oocytes with PB1 of a homogeneous, round shape with a smooth or intact surface may be indicative of a better oocyte, the usefulness of this selection criterion in livestock requires further research to establish its real predictive value for oocyte competence.

### 2.7. Polarized Light Microscopy

Polarized light microscopy (PLM) has been used in different mammalian oocytes since it allows the noninvasive assessment of subcellular features such as the meiotic spindle and zona pellucida birefringence (ZPB). To learn about the principles and equipment required for PLM in detail, readers are directed to excellent reviews on the subject [105,106].

#### 2.7.1. Evaluation of the Meiotic Spindle and Zona Pellucida Birefringence

Using PLM, it is possible to locate and evaluate the morphology of the meiotic spindle to confirm egg maturation, which has been positively correlated with developmental competence [90,107,108,109]. This method avoids damaging the spindle during the ICSI procedure, considering that the position of the PB1 can be altered when CCs are removed during preparation for ICSI [110]. Furthermore, PLM has been successfully used to remove the meiotic spindle and chromosomes (enucleation) in mice [111], bovines [112], and pigs [113], with an average efficiency of 90% and, more importantly, avoiding the exposure to ultraviolet (UV) rays and their detrimental effect on embryonic development.

In livestock species, the dark appearance of the ooplasm, attributed to high lipid contents, is known to interfere with spindle imaging [113] and, as in humans, precludes the detection of meiotic spindle abnormalities [102,113,114]. Therefore, spindle birefringence should be carefully considered as an index of gamete quality and chromosome alignment in some species. In pigs, a negative PLM signal was associated with to reduced maturation and poor development potential [113]. In the same study, when the PLM system was used for spindle removal, the overall enucleation efficiency was 92.6%, indicating that PLM is an effective tool for performing enucleation in pigs. A few years later, the same group evaluated the use of PLM to assess the meiotic spindle of in vitro matured bovine oocytes after vitrification and warming [115]. They were able to confirm the presence of the meiotic spindle in 99% of the analyzed eggs. Moreover, after vitrification and warming, meiotic spindles were detected in 79% of oocytes. Interestingly, thawed oocytes that displayed a positive PLM signal showed better competence in terms of cleavage and blastocyst rates after parthenogenetic activation, indicating that PLM can be a useful tool for assessing post-warming viability in vitrified bovine oocytes.

Overall, these studies demonstrate that PLM efficiently detects the meiotic spindle of livestock oocytes and does not affect early embryonic development. However, the selection of cattle oocytes on the basis of the presence of a PLM signal does not seem to offer improvement in IVP outcomes yet.

#### 2.7.2. Assessment of the Zona Pellucida Birefringence

In addition, PLM has been used for the evaluation of the ZPB, which in humans has been associated with oocyte quality [116,117,118], although this is still under debate [119,120]. The few studies in cattle showed that a lower ZPB is related to high-quality oocytes and improved blastocyst development [121,122], whereas two studies in horses reported conflicting results, indicating beneficial effects of both low ZPB [123] and high ZPB [124]. Because most of the studies with PLM were carried out in mice and humans with conflicting results, its potential application and practical use in cattle and other livestock species needs further assessment. Contrary to humans, where the number of highly valuable oocytes from donors is relatively low, livestock oocytes obtained from slaughterhouse ovaries allow a more stringent selection. Furthermore, assessment of the meiotic spindle can be a laborious procedure, which delays the overall process of in vitro manipulation and embryo production. Thus, its application will require showing a clear advantage over conventional approaches using the morphological criterion mentioned above for oocyte selection. However, PLM might be beneficial when individual oocytes are of high value, such as oocytes recovered from elite cows by ovum pick-up (OPU) [111,113].

### 2.8. Brilliant Cresyl Blue (BCB) Staining

Another approach that demonstrated predictive potential is the evaluation of glucose-6-phosphate dehydrogenase (G6PDH) activity via brilliant cresyl blue (BCB) staining. BCB is a dye that determines the intracellular activity of G6PDH. Activity of G6PDH is observed during the oocyte growth phase (BCB^−^: colorless cytoplasm, increased G6PDH) due to the demand of ribose-6-phosphate for nucleotide synthesis. This activity is low (BCB^+^: colored cytoplasm, low G6PDH) in oocytes that have completed their growth phase [125]. This technique has been successfully employed in various species, including cattle [125,126,127].

Although previous reports found that the developmental competence of oocytes with low G6PDH activity (BCB^+^) was higher than that of oocytes with a high G6PDH activity (BCB^−^), the absence of differences in terms of embryonic development between BCB^+^ and the untreated control group decreases the utility of the BCB test in IVP technology [128]. However, it is unquestionable that BCB^+^ oocytes have statistically higher developmental competence than BCB^−^ oocytes, both in IVF and somatic cell nuclear transfer (SCNT) [128].

Later studies continued to show only a trend of BCB^+^ oocytes toward greater developmental potential. Better blastocyst rates at day 7 were reported by Silva et al. [129], and a study by Fakruzzaman et al. [130] reported higher blastocyst quality on the basis of total apoptotic cells and mitochondria numbers. Similarly, Castaneda et al. [131] indicated that the higher lipid content of BCB^+^ bovine oocytes might be associated with their better developmental competence. Interestingly, another article indicated that co-culture with BCB^−^ oocytes during IVM affects negatively the capacity of BCB^+^ oocytes to undergo embryonic development [132]. However, other authors suggested that the BCB test is not sufficient for identification of the most competent gametes [133]. Nonetheless, the combination of oocyte and zygote selection using BCB staining would improve the efficiency of embryo selection [134]. Therefore, the BCB test can be a valuable tool when used together with classical morphological classification and could be useful for the selection of oocytes with a higher implantation potential. Nonetheless, an assessment of the effects of BCB staining on post-implantation development is necessary to elucidate its usefulness for IVP technologies, not only for research but also in the industry of animal production. A summary of the morphological and visual indicators associated with oocyte competence is shown in Table 1.

## 3. Non-invasive Molecular Approaches

Many studies are being performed in mammals in order to find molecular markers predictive of oocyte quality. So far, most of the data show considerable variations, perhaps due to different experimental conditions and/or the criterion of quality/competence, resulting in varied scientific views.

### 3.1. Cell Death (Apoptosis) in Cumulus Cells

Because morphological evaluation prior to maturation does not allow to discriminate the atretic oocytes from healthier ones [135], one of the earlier noninvasive markers of oocyte competence was the level of apoptosis in CC, seen as DNA fragmentation, externalization of phosphatidylserine (EP), and/or the expression ratio of anti-apoptotic (Bcl-2) and pro-apoptotic (Bax) genes (BCL-2/BAX). Early studies found that the CC of bovine COCs undergo progressive apoptosis during IVM [136], and this was negatively correlated with the oocyte developmental capacity [51]. However, results reported by Janowski et al. [137] supported the notion that follicular cells surrounding the more competent oocytes have a higher degree of apoptosis. Later, Warzych et al. [138] showed that the level of apoptosis in CC was not associated with morphology or the oocyte meiotic stage, suggesting that the extent of apoptosis in CC is not a reliable quality marker for gamete competence. Similarly, the study of Anguita et al. [135] showed that embryonic developmental potential increased together with oocyte diameter, but this developmental competence was not related to the incidence of apoptosis. Recently, another study indicated that optimum control of the meiosis block, nuclear maturation, and developmental potential were associated with less DNA fragmentation in CC [50].

Similarly, in the human model, the majority of related studies have focused on granulosa cells (GC) isolated from FF during oocyte collection. Apoptosis, evaluated by EP, of GC was negatively associated with egg and embryo numbers in IVF/ICSI cycles, pregnancy rate, and live birth rate after IVF [139,140]. However, contrarily, it was also reported that the EP in GC is not related to follicular quality and oocyte competence during ICSI [141]. Thus, in the bovine and human models, it is still controversial whether apoptosis of GC and/or CC can impact the developmental potential of the oocyte.

### 3.2. Transcriptomic and Proteomic of Cumulus Cells

Many new genomic tools helped to deepen the understanding in the area of oocyte–cumulus communication, as well as molecular pathways required for the acquisition of competence in mammalian gametes and embryos. For instance, recent advances in RNA-Seq technology offer a global transcriptomic approach for identifying differentially expressed genes associated with competence and embryonic development.

Among the molecular approaches, study of the transcriptomic profile of the surrounding cumulus is one of the most popular attempts at finding molecular markers associated with gamete competence in mammals. The “noninvasive” strategy is based on profiling the gene expression of a small biopsy before IVM, maintaining COC integrity, and following the embryonic development of the respective oocyte. This is also called “oocyte fate” [142]. Although several studies in cattle already found several genes in CCs from germinal vesicle (GV) [16,35,143,144,145,146,147,148,149,150,151] and MII oocytes [144,152] to be associated with oocyte competence, only a few reports matched the oocyte fate with the transcriptomic profile obtained from the CCs or granulosa cells (Table 2). There is some consensus regarding pathways correlated positively with oocyte competence, including the cell cycle (CCND1, CCNB2, and CCNA2 genes) [143,145,153], cell growth and proliferation, (CD44, TGFB1, EGF, FGF11, PRL, and GH genes) [35,147,148,149,154], and steroidogenesis (HSD3B2 and CYP11A1 genes) [16,154]. On the contrary, genes related to cellular apoptosis would be associated with a low competence (ATRX, KRT8, ANGPT2, KCNJ8, and ANKRD1 genes) [142,147,152,155].

On the other hand, studies analyzing the proteomic profile of the cumulus–oocyte complex (COC) are scarce. Moreover, most of them have done invasive analysis in a pool of oocytes; thus, oocyte fate could not be followed (Table 2). Nonetheless, the few studies described many proteins involved in cell signaling that may have a role in cumulus–oocyte communication and competence. Most of the proteins are involved in components of integrin, actin cytoskeleton, mitogen-activated protein kinases (MAPK) and phosphatidylinositol 3-kinase (PI3K) signaling pathways, extracellular matrix (ECM) receptor interactions, steroid biosynthesis, and glucose and carbohydrate metabolism, which may have implications in various reproductive processes such as oocyte development and maturation [156,157,158] (Table 2). A recent study reported a highly sensitive approach to characterize the CC proteome from a single COC after in vivo or in vitro maturation [156]. This method shows the potential to directly connect the cumulus proteome to the developmental potential of the corresponding oocyte, as already performed at the gene expression level.

### 3.3. Follicular Fluid Analysis

It is well known that the composition of FF has an impact on the developmental capacity of the oocyte and, thus, the resulting embryo. Excellent articles reviewed the importance of FF on oocyte physiology and fertility [159,160,161]. This fluid contains proteins, cytokines, growth factors, steroids, metabolites, and other indeterminate factors [159]. Therefore, by studying its composition, it should be possible to predict oocyte competence and fertilization outcomes [162,163,164]. Metabolites in the FF, such as glucose and potassium, have already been positively associated with oocyte quality in cattle [41,165]. However, studies linking the FF features with the respective oocyte fate in bovines have not been performed yet. Reports in humans have positively associated the presence of anti-Müllerian hormone (AMH) in FF with competence of the respective oocyte [166,167], although with some contradictory results [168,169]. Conversely, a recent study that used a large population of transferred embryos matching FF samples indicated that the AMH level in FF following withdrawal from the ovarian follicle is closely linked to the oocyte’s competence, and it is a suitable predictor of a live birth after single embryo transfer [170]. In the cow, it was already reported that AMH concentrations can be predictive of the number of ovulations and embryos produced in response to ovarian stimulation by FSH [171,172,173], making it a suitable molecule to be related to the oocyte competence.

In addition, other molecules in FF of cattle that show promising results are microRNAs (miRNAs). The bovine FF contains free miRNAs, as well as some associated with exosomes [174,175]. Recently, the study of Pasquariello et al. [176] showed, for the first time, the miRNA content of different populations of oocytes categorized according to their competence. Interestingly, they discovered that the most differentially expressed miRNAs (miR-24, miR-10a, and miR-320a) in FF found in highly competent follicles were part of the regulation of the neurotrophin signaling pathway, which supports follicle formation and development, as well as the TGF-βsignaling pathway that controls the production of ovarian peptide hormones. Therefore, linking FF molecules such as AMH or miRNAs with gamete competence is an encouraging strategy in the field of oocyte selection. However, we have to consider that it will be applicable only when the fast collection and analysis of FF from individual follicles become practicable.

## 4. Conclusions and Future Perspectives

The classification and selection of oocytes in livestock species for in vitro embryo production and for micromanipulation techniques, such as ICSI and SCNT, can be one of the most important steps to reach superior embryonic development and quality. Although more sophisticated methods (qRT-PCR, global transcriptomic, and proteomic analysis) have been studied since a few decades ago, the lack of a quick enough method producing reliable results hinders the implementation of these technologies. Moreover, molecular analysis requires high-tech equipment and technical staff that would be cost-ineffective in most research laboratories. Thus, although oocyte selection based on morphologic criteria appears to be insufficient to distinguish more competent gametes, in real practice, when 100–300 oocytes are waiting to be processed during micromanipulation experiments, it seems to be the only available strategy so far. Furthermore, studies that perform embryo transfers are also important to effectively evaluate developmental potential, as successful embryo implantation is highly dependent on the quality of the embryo and the intricate relationship it establishes with the uterine endometrium. Ultimately, with the advent of bovine embryonic stem cells, greater scrutiny of oocytes with high developmental potential is necessary, for the production of stable pluripotent stem cell lines to be used in basic science, forward and reverse genetics, epigenetics, gene imprinting, and the production of animal models with applications in animal production. Thus, in addition to improving the conditions to support in vitro maturation, the implementation of new tools for the assessment of gamete competence, together with studies decoding molecular cues in oocyte maturation, will improve our understanding of this complex process and will more precisely identify the synchrony between nuclear and cytoplasmic maturation in livestock species.

## Figures and Tables

**Figure 1 animals-10-02196-f001:**
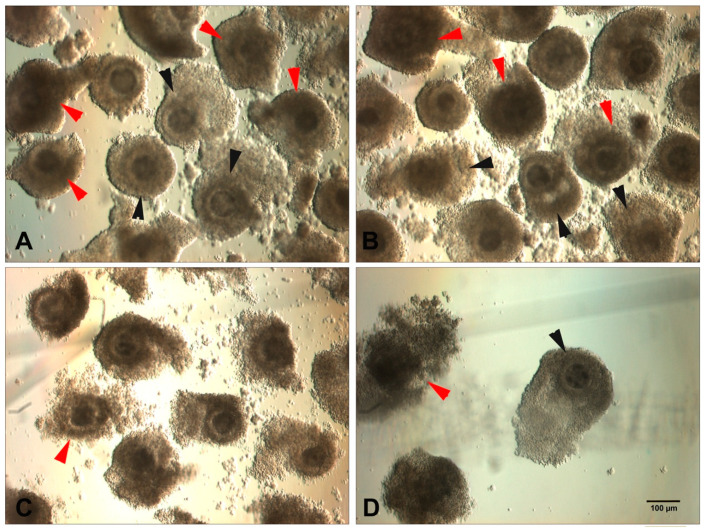
Representative images of bovine cumulus–oocyte complexes (COCs) after ovary withdrawal classified according to COC morphology. (**A**,**B**) Complete cumulus cover with several compacts (red arrows) and slightly loose (black arrows) cell layers; (**C**) partial cumulus cover and loose cell layers with signs of early atresia (red arrow); (**D**) COC showing clear signs of atresia (red arrow) and a black-punctate cytoplasm (black arrow).

**Figure 2 animals-10-02196-f002:**
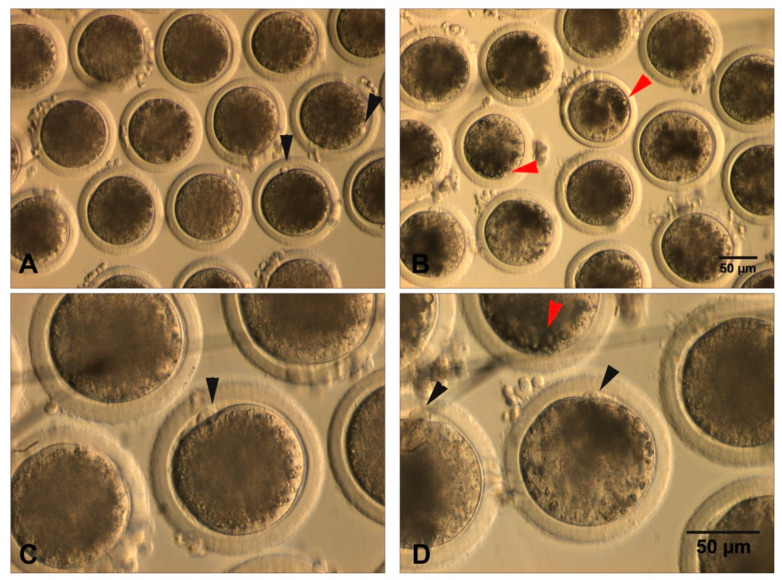
Denuded MII bovine oocytes after 24 h of IVM. (**A**,**C**) oocytes showing a homogeneous dark cytoplasm. Black arrows depict the first polar body; (**B**,**D**) oocytes showing a heterogeneous pale and punctuated cytoplasm. Black arrows indicate the first polar body, while red arrows depict dark areas of intense lipid accumulation (cytoplasmic granulations).

**Table 1 animals-10-02196-t001:** Summary of the morphological and visual indicators of oocyte competence.

Reference	Criteria	Recommendation
[28,30,31]	Ovarian morphology	Presence of cycle-related structures
[7,33,34,35]	Follicle size	>5 mm
[30,49,50]	Morphology of the cumulus–oocyte complexes (COCs)	COCs with at least five layers of cumulus cells (CC), compact and/or slightly expanded cumulus, with or without dark spots in the oocyte and cumulus
[53,67,68]	Lipid content	Dark ooplasm indicates high competence, light-colored indicates lacking lipids and poor competence, and black ooplasm indicates aging
[77,78,79,80]	Cumulus expansion and oocyte size	Not associated to oocyte quality; important role in fertilization
[29,83,86,135]	Oocyte size	Diameters >115 and <130 microns
[96,97,98,99]	First polar body (PB1) morphology	PB1 of a homogeneous, round shape with a smooth or intact surface
[112,114,115]	Meiotic spindle and zona pellucida birefringence	Useful tool for micromanipulation procedures (intracytoplasmic sperm injection (ICSI) or somatic cell nuclear transfer (SCNT)) and for assessing post-warming integrity of meiotic spindle of vitrified bovine oocytes
[121,122]	Zona pellucida birefringence (ZPB)	Lower ZPB is related to high quality oocytes and improved blastocyst development
[115,128,129,130,134]	Brilliant cresyl blue staining	BCB^+^ oocytes have higher developmental competence than BCB^−^ oocytes

**Table 2 animals-10-02196-t002:** Summary of studies performing transcriptomic and proteomic analysis of CC and/or GC.

Reference	Oocyte Stage	Criterion of Developmental Competence	Technique Used	Genes and/or Pathways Associated with High Competence	Genes and/or Pathways Associated with Low Competence
**Transcriptomic**					
[16]	GV	GC collected 2 h before and 6 h after LH surge	qPCR and microarray analysis	TNFAIP6, HAS2, HSD3B2, PLOD2, CHSY1 (differentiation, cell growth, protein translation, apoptosis-related, lipid and glucose metabolism, ECM formation)	ENO1, DNAJB6, GJA1, SYNPO, ZNF330, MYO1D (protein synthesis cellular movement, cell signaling, molecular transport, nucleic acid metabolism)
[35]	GV	follicle size (1.0–3.0, 3.1–6.0, 6.1–8.0, and ≥8.1 mm)	qPCR	FSHR (follicle stimulant hormone receptor), GH (cell growth), and EGF (cell growth and differentiation)	N.A
[142]	GV	Cell arrest and oocyte fate	qPCR and microarray analysis	GATM (post-translational modification, amino-acid metabolism, and free-radical scavenging)	AGPAT9 (lipid metabolism), CLIC3 (chloride ion concentration control, cell volume regulation, and apoptosis), KRT8 (cellular assembly and organization, apoptosis)
[143]	GV	Follicle size (>5 mm vs. <2mm)	qPCR and SSH	Oct4, Msx1 (transcription factors), Znf198 (TFGb and activin signaling), NDFIP1(posttranslational modification), CCNA2 (cell cycle), SLB (stabilization and translation of mRNAs encoding histones)	N.A
[144]	GV	Adult vs. prepuberal donors	qPCR and microarray analysis	N.A	CTSB, CTSK, CTSS, and CTSZ (cathepsin family of lysosomal cysteine proteinases)
[145]	GV	OPU 6 h post LH vs. slaughterhouse oocytes after 6 h IVM	qPCR and microarray analysis	PTTG1, CDC5L, CKS1B, CCNB2 (cell cycle), PSMB2, PRDX1 (cell metabolism), RGS16 (cell signaling), SKIIP (gene expression), and chromatin support H2A	BMP15, GDF9, CCNB1, and STK6 (follicle–oocyte interaction and cell cycle)
[146]	GV	Brilliant cresyl blue staining	qPCR	N.A	CTSB, CTSK, CTSS, and CTSZ (cathepsin family of lysosomal cysteine proteinases)
[147]	GV	GC collected after FSH withdrawal	qPCR and microarray analysis	SMAD7, STAT1 (transcription), PRL and GH (cell growth, proliferation), BMPR1B, IGF2, RELN, and TFPI2 (follicle growth), NRP1 (angiogenesis), GFPT2, TF, and VNN1 (oxidative stress response)	KCNJ8 and ANKRD1 (apoptosis and inflammation)
[148]	GV	Follicle size (>8 mm vs. <3mm)	qPCR and microarray analysis	FGF11 (cell growth, and differentiation), IGFBP4 and SPRY1 (cell cycle, DNA repair)	ARHGAP22, COL18A1, and GPC4 (cell cycle, signaling)
[149]	GV	IVM plus FSH or phorbol myristate acetate (PMA) treatment	qPCR and microarray analysis	HAS2, INHBA, EGFR, GREM1, CD44, TNFAIP6, PTGS2, HSP90B1, SERPINE2, PTX3 (differentiation, cell growth, protein translation, apoptosis, lipid and glucose metabolism, ECM formation)	N.A
[150]	GV	Follicle size and oocyte fate	qPCR	GPC4 (regulation of growth factors, adhesion, signaling, proliferation, and differentiation)	N.A
[151]	GV	COCs morphology and oocyte fate	qPCR	N.A	FSHR, IGF1R, CYP11al, and HSD3β (cell growth, cell differentiation, steroidogenesis)
[153]	GV	Maturation outcome and oocyte fate	RNA-seq	CCND1, BMP15, GDF9, H19, KLF4, GPC1, SYCP3, and CTSB (cell cycle, meiosis, cell signaling, metabolism, and apoptosis)	N.A
[154]	GV	FSH withdrawals; follicles from 5 mm aspirated by OPU	qPCR and microarray analysis	CYP11A1 (steroidogenesis), NSDHL (cholesterol synthesis), GATM (creatine biosynthesis), MAN1A1 (functional gap junction-mediated communication), VNN1 (oxidative stress response), NRP1 (angiogenesis), TGFB1 (cell growth and differentiation)	N.A
[155]	GV	Chromatin compaction, follicle size, and BCB staining	qPCR and microarray analysis	GATM (posttranslational modification, amino-acid metabolism, and free-radical scavenging), MAN1A1 (functional gap junction-mediated COC communication), ZIP8 (zinc transporter)	ANGPT2 (cell death, apoptosis)
**Proteomic**					
[156]	MII	Matured in vivo vs. IVM	MALDI TOF	KEGG pathways of the complement and coagulation cascade, ECM–receptor interactions, steroid biosynthesis, glucose and carbohydrate metabolism	N.A
[157]	GV	COC morphology and follicle size (>2 mm to 8 mm)	2-DLCMS	Integrin signaling, actin cytoskeleton signaling, ephrin receptor signaling, PI3K signaling, MAPK signaling	N.A
[158]	GV	COC morphology and follicle size (>2 mm to 8 mm)	2-DLCMS	4395 proteins were expressed in the CCs; 858 proteins were common to both CCs and oocytes	N.A

MII: meiosis II, GV: germinal vesicle, CC: cumulus cells, GC: granulosa cells, qPCR: quantitative reverse transcription PCR, RNA-seq: RNA sequencing, IF: immunofluorescence, SSH: suppressive subtractive hybridization, BCB: Brilliant cresyl blue, 2-DLCMS: two-dimensional liquid chromatography-tandem mass spectrometry, MALDI TOF: matrix-assisted laser desorption/ionization-time of flight, ECM: extracellular matrix, MAPK: mitogen-activated protein kinases, PI3K: phosphatidylinositol 3-kinase, IVM: in vitro *maturation*. * N.A = not available.

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
