# Peer review of "Oocyte Selection for In Vitro Embryo Production in Bovine Species: Noninvasive Approaches for New Challenges of Oocyte Competence"

_animals, 2020, doi:10.3390/ani10122196_

Round 1
Reviewer 1 Report
The review is well written and provides comprehensive description of the selected topics. Unfortunately, the authors concentrated their attention only on very simple techniques for assessing oocyte quality, largely based on eye or microscopic evaluation of ovaries, oocytes and cumulus cells. Such an approach has been applied in number of previous reviews without clear conclusions for research applications and practice. Considering the increasing availability of molecular methods to the studies of the oocytes, the review appears somewhat old-fashioned. I suggest that expansion of the topics of this review by non-invasive molecular methods used for assessment of oocyte quality, like expression of specific genes in cumulus cells, proteomic analysis of cumulus cells, detection of specific miRNA in cumulus cells, follicular fluid and culture media, ... would strengthen the impact of this work and better fulfil expectations of readers.
Specific comments:
Line 98: split “inone” to “in one”
Line 104: delete “that”
Line 353: do not repeat definition of glucose-6-phosphate dehydrogenase, use only abbreviation G6PDH here
Author Response
Point 1. Considering the increasing availability of molecular methods to the studies of the oocytes, the review appears somewhat old-fashioned. I suggest that expansion of the topics of this review by non-invasive molecular methods used for assessment of oocyte quality, like expression of specific genes in cumulus cells, proteomic analysis of cumulus cells, detection of specific miRNA in cumulus cells, follicular fluid and culture media, ... would strengthen the impact of this work and better fulfil expectations of readers.
Response 1. We would like to thank the referee for all the constructive comments that have improved the manuscript. We agreed with the reviewer and we have included a new section of non-invasive molecular methods.
Point 2. Specific comments:
Line 98: split “inone” to “in one”
Line 104: delete “that”
Line 353: do not repeat definition of glucose-6-phosphate dehydrogenase, use only abbreviation G6PDH here
Response 2. All specific comments have been corrected according to the reviewer’s suggestions.
Reviewer 2 Report
The title of the manuscript is not adequate to the contents. It erroneously suggests that the work will concern various species of farm animals but is focused only on cattle.
The information presented in this paper are generally known for many years and I do not see "new chelenges" and an innovative approach to the subject.
On the other side, the article contains a well-done review of the literature and is correctly written in terms of content and logic.
Unfortunately, in my opinion, the manuscript in the present form is not acceptable for publication in journal with 2,3 IF.
The manuscript should be revised. Assisted reproduction techiques in cattle are very effective in compare to other livestock species (horses, pigs). I suggests adding a comparison with other species e.g. the specific structure of the mare's ovary, different oocytes classification systems, different conditions and efficiency of IVM, differences in the lipids content and distribution etc. These changes will make the article more interesting.
Author Response
Point 1. The title of the manuscript is not adequate to the contents. It erroneously suggests that the work will concern various species of farm animals but is focused only on cattle.
Response 1. We would like to thank the referee for all the constructive comments that have improved the manuscript. We are in agreement with the reviewer comments and we have modified the title making it more specific for bovine species
Point 2. The information presented in this paper are generally known for many years and I do not see "new chelenges" and an innovative approach to the subject.
Response 2. Line 77 to 82 we have included a brief description regarding why there is a needing for choosing the best oocytes for IVP in cattle.
Point 3. The manuscript should be revised. Assisted reproduction techniques in cattle are very effective in comparison to other livestock species (horses, pigs). I suggests adding a comparison with other species e.g. the specific structure of the mare's ovary, different oocytes classification systems, different conditions and efficiency of IVM, differences in the lipids content and distribution etc. These changes will make the article more interesting.
Response 2. We are in agreement with the reviewer's comments and we have included a new section about non-invasive molecular methods.
Reviewer 3 Report
In the present article a review is performed related with the oocyte competence and its evaluation to select for the improvement of the in vitro embryo production. This subject is relevant due to the low efficiency of the blastocyst production during in vitro culture.
The authors make a deep review of the literature and the article is well organized and well written. I recommend its aceptation after some changes that I consider that will improve the review.
MAJOR COMMENTS
1.- I strongly recommend to include some tables with the major findings and the references. This will be very useful for the readers.
2.- I strongly recommend to include some relevant images about the oocyte morphology, expanded cumulus,
3.- I consider that it is important to include relevant studies about the follicular fluid and gene expression related with the oocyte maturation. Please provide more information (Lines 133-140).
4.- Please include studies about the genomic analysis of the polar body as a non-invasive procedure for oocyte selection.
MINOR COMMENTS
1.- Lines 1, 34. Please check in vitro in the document. Italic.
2.- Line 29. I consider that the diameter of the oocyte of 120 microns should not be used because the authors are talking about mammalian species in general.
3.- Lines 65-66. Probably the main difference is not the ovarian follicle maturity but the metaphase II status. Please revised it.
4.- Line 98. Please revise the "inone" spelling.
5.- About lipid droplets, I recommend to include relevant data about the following article or similar Live-cell quantification and comparison of mammalian oocyte cytosolic lipid content between species, during development, and in relation to body composition using nonlinear vibrational microscopy.
6.- Lines 133-220. Please provide the reference.
7.- Line 229. "..a low density of organelles....". It is better to be more precise. Organelles is probably to wide.
8.- Line 270. 2.6. First polar body assessment.
I recommend to include relevant information about the molecular analysis of the PB because it is a non-invasive methodology previously used.
9.- Line 306. I consider that this section can be subdivided in two different one about the meiotic spindle and another one for the zona pellucida.
10.- Lines 335-336. More recent relevant references should be included.
11.- Lines 370-372. This sentence is not clear.
12.- Lines 373-374. I consider that this sentence is not necessary. It is evident that the oocyte maturation is a complex process and not only the G6PDH is involved in this process.
13.- Lines 374-376. I consider that posterior embryos stage would be also important or relevant specially before the genome activation.
Author Response
Point 1. I strongly recommend to include some tables with the major findings and the references. This will be very useful for the readers. I strongly recommend to include some relevant images about the oocyte morphology, expanded cumulus
Response 1. We would like to thank the referee for all the constructive comments that have improved the manuscript. We are in agreement with the reviewer comments and we have included tables summarizing major findings and the references, as well as representative images about oocyte’s morphologies.
Point 2. I consider that it is important to include relevant studies about the follicular fluid and gene expression related with the oocyte maturation. Please provide more information (Lines 133-140).
Response 2. We are in agreement with the reviewer comments and now it is included a new section of non-invasive molecular methods.
Point 3. Please include studies about the genomic analysis of the polar body as a non-invasive procedure for oocyte selection.
Response 3. We are in agreement with the reviewer comments, however, because polar body biopsy has evolved into a secure method to assess the chromosomal status of the oocyte, and due to the biopsy of both polar bodies (PBs) has to be performed after fertilization, we consider this technique as a method to select embryos (''zygotes'') more than a tool to choose oocytes for IVP. (van der Ven K, Montag M, van der Ven H. Polar body diagnosis a step in the right direction? doi:10.3238/arztebl.2008.0190). Thus, we have considered it out of bounds for this review.
Point 4. MINOR COMMENTS
1.- Lines 1, 34. Please check in vitro in the document. Italic.
2.- Line 29. I consider that the diameter of the oocyte of 120 microns should not be used because the authors are talking about mammalian species in general.
3.- Lines 65-66. Probably the main difference is not the ovarian follicle maturity but the metaphase II status. Please revised it.
4.- Line 98. Please revise the "inone" spelling.
5.- About lipid droplets, I recommend to include relevant data about the following article or similar Live-cell quantification and comparison of mammalian oocyte cytosolic lipid content between species, during development, and in relation to body composition using nonlinear vibrational microscopy. Joshua Jasensky, Andrew P Boughton, Alexander Khmaladze, Jun Ding, Chi Zhang, Jason E Swain, George W Smith, Zhan Chen, Gary D Smith. 2016 Aug 7;141(15):4694-706. doi: 10.1039/c6an00629a.
6.- Lines 133-220. Please provide the reference.
7.- Line 229. "..a low density of organelles....". It is better to be more precise. Organelles is probably to wide.
8.- Line 270. 2.6. First polar body assessment. I recommend to include relevant information about the molecular analysis of the PB because it is a non-invasive methodology previously used.
9.- Line 306. I consider that this section can be subdivided in two different one about the meiotic spindle and another one for the zona pellucida.
10.- Lines 335-336. More recent relevant references should be included.
11.- Lines 370-372. This sentence is not clear.
12.- Lines 373-374. I consider that this sentence is not necessary. It is evident that the oocyte maturation is a complex process and not only the G6PDH is involved in this process.
13.- Lines 374-376. I consider that posterior embryos stage would be also important or relevant specially before the genome activation.
Response 4. All the minor comments were corrected according to the reviewer’s suggestions (Except for point 8, since we consider this technique as a method to select zygotes, as previously stated)
Round 2
Reviewer 1 Report
The authors responded to all of my comments and revised the manuscript accordingly. The version 2 provides comprehensive description of the selected topic and does not lack author´s critical evaluation of the previous work. I consider the revised manuscript suitable for publication in Animals.
Reviewer 2 Report
The manuscript has been revised according to the reviewers' suggestions and became more interesting. A new table and section about non-invasive molecular methods increased the attractiveness of the paper. In this form I recommend the manuscript for publication.